# Local Reinforcement Learning with Action-Conditioned Root Mean Squared Q-Functions

**Frank Wu**
Carnegie Mellon University
Pittsburgh, PA, USA
`frankwu2@cs.cmu.edu`

**Mengye Ren**
New York University
New York, NY, USA
`mengye@nyu.edu`

## Abstract

The Forward-Forward (FF) Algorithm is a recently proposed learning procedure for neural networks that employs two forward passes instead of the traditional forward and backward passes used in backpropagation. However, FF remains largely confined to supervised settings, leaving a gap at domains where learning signals can be yielded more naturally such as RL. In this work, inspired by FF's goodness function using layer activity statistics, we introduce Action-conditioned Root mean squared Q-Functions (ARQ), a novel value estimation method that applies a goodness function and action conditioning for local RL using temporal difference learning. Despite its simplicity and biological grounding, our approach achieves superior performance compared to state-of-the-art local backprop-free RL methods in the MinAtar and the DeepMind Control Suite benchmarks, while also outperforming algorithms trained with backpropagation on most tasks.

## 1 Introduction

The success of deep learning has relied on *backpropagation* (Rumelhart et al., 1986), a procedure that has significant limitations in terms of biological plausibility as it requires synchronous computations and weight symmetry. Many works have provided backprop-free alternatives for training deep neural networks (Lillicrap et al., 2016; Nøkland, 2016; Nøkland & Eidnes, 2019). Notably, Hinton (2022) proposed the Forward-Forward algorithm (FF), a new approach that performs layerwise contrastive learning between positive and negative samples. This algorithm is lightweight and entirely eliminates the need for backpropagation, thereby addressing some of the biological plausibility concerns.

However, most studies on backprop-free methods are focused on the search for a biologically plausible mechanism for performing gradient updates on supervised tasks. Could a biologically plausible source of learning signals be equally meaningful? Reward-centric environments and temporal-difference (TD) methods (Sutton, 1988) serve as natural candidates for filling this gap. Biological brains have evolved through a series of reward-guided evolution, while ample evidence has shown that our brains could be implementing TD (Schultz et al., 1997a; O'Doherty et al., 2003; Watabe-Uchida et al., 2017; Amo et al., 2022). Since the goodness score in FF models the "compatibility" between the inputs and labels, this local learning paradigm can be readily adapted to a reinforcement learning (RL) setting where we model the value of an input state and an action from each layer's activities. See Figure 1 for a comparison between the supervised learning and RL setups of the forward-forward learning paradigm.

Towards integrating local methods and RL, Guan et al. (2024) recently proposed Artificial Dopamine (AD) that incorporates top-down and temporal connections in an Q-learning framework. Since the local Q-Function estimation needs to be explicitly predicted, Guan et al. (2024) uses a dot-product between two sets of mappings from the inputs that produces the value estimate for each action. This design, while backprop-free, makes the architecture more flexible modeling complex inputs. However, AD still relies on the output of the dot-product to be the same dimension as the action space, limiting the performance of the method.

Inspired by FF's local goodness function from using layer statistics, we propose Action-conditioned Root mean squared Q-Function (ARQ), a simple vector-based alternative to traditional scalar-based

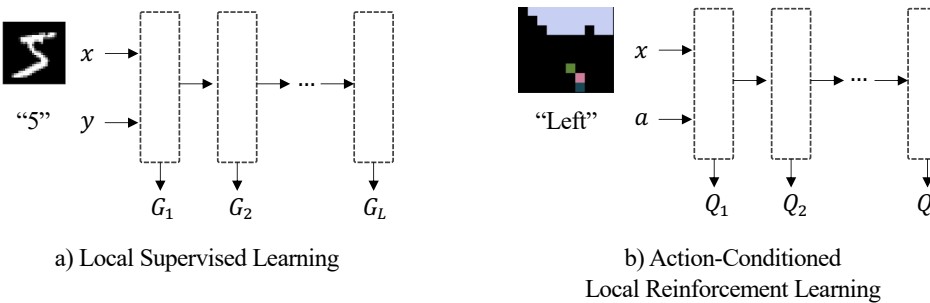

a) Local Supervised Learning          b) Action-Conditioned
Local Reinforcement Learning

Figure 1: Local learning paradigms inspired by the Forward-Forward (FF) algorithm (Hinton, 2022). a) The original FF is designed for supervised learning, where each layer models the "goodness" between image $x$ and label $y$. Information is carried forward only through bottom-up and optionally top-down connections without backpropagation. b) We extend FF local learning for reinforcement learning—each layer takes a state observation $x$ and an action candidate $a$ as input, and estimates the Q value by taking the root mean squared function of the hidden vector.

Q-value predictors designed for local RL. ARQ is composed of two key ingredients: a goodness function that extracts value predictions from a vector of arbitrary size, and action conditioning by inserting an action candidate at the model input. ARQ significantly improves the expressivity of a local cell by allowing more neurons at the output layer without sacrificing the backprop-free property. By applying action conditioning, we further unleash the capacity of the network to produce representation specific to each state-action pair. Moreover, ARQ can be readily implemented on AD and take full advantage of their non-linearity and attention-like mechanisms.

We evaluate our method on the MinAtar benchmark (Young & Tian, 2019) and the DeepMind Control Suite, challenging suites designed to test RL algorithms in low-dimensional settings where local methods remain viable. Our results show that our method consistently outperforms current local RL methods and surpasses conventional backprop-based value-learning methods in most games, demonstrating strong decision-making capabilities without relying on backpropagation. Through this contribution, we seek to encourage further exploration of the intersection between RL and biologically plausible learning methods.

## 2 RELATED WORKS

**Backprop-free learning methods & FF:** In recent years, several backprop-free training algorithms have been proposed to address the limitations of traditional backpropagation in neural networks (Lillicrap et al., 2016; Nøkland, 2016; Nøkland & Eidnes, 2019; Belilovsky et al., 2019; Baydin et al., 2022; Ren et al., 2023; Fournier et al., 2023; Singhal et al., 2023; Innocenti et al., 2025). One notable method is the Forward-Forward Algorithm (FF) (Hinton, 2022), which offers a biologically plausible, energy-efficient alternative to backpropagation. To extend the capabilities of FF, Ororbia & Mali (2023) proposed the Predictive Forward-Forward Algorithm, showing that a top-down generative circuit can be trained jointly with FF. Tosato et al. (2023) found that models trained with FF objectives generate highly sparse representations. This pattern closely resembles the observations of neuronal ensembles in cortical sensory areas, suggesting FF may be a suitable candidate for modeling biological learning. Recently, Sun et al. (2025) proposed DeeperForward, integrating residual connections (He et al., 2016), the mean goodness function, and a channel-wise cross-entropy based objective function (Papachristodoulou et al., 2024) into FF. DeeperForward yields 87% on CIFAR-10 with a 17-layer deep architecture.

**Value Estimation in Deep Neural Networks:** TD methods for value estimation have been particularly useful in the recent decade as the rise of deep neural networks offers a powerful function approximator. Mnih et al. (2013) introduced DQN, where a deep neural network is applied to approximate the Q-Function. They showed that this method significantly outperformed earlier methods on the Atari 2600 games, initiating a family of methods built upon this architecture (Van Hasselt et al., 2016; Wang et al., 2016; Dabney et al., 2018b; Hessel et al., 2018; Fortunato et al., 2017; Dabney et al., 2018a; Hausknecht & Stone, 2015). In actor-critic architectures, it is also common to use a deep neural network for value and advantage estimation (Schulman et al., 2017; 2015a;b; Lillicrap

et al., 2015; Mnih et al., 2016; Haarnoja et al., 2018b;a; Fujimoto et al., 2018; Gruslys et al., 2017; Abdolmaleki et al., 2018; Kostrikov et al., 2020; Yarats et al., 2021). For planning-based methods using either Monte Carlo tree search (MCTS) or a learned model, value estimation is also significant in driving the planning process (Schrittwieser et al., 2020; Silver et al., 2016; 2017; Hansen et al., 2023; Sacks et al., 2024; Ye et al., 2021; Hafner et al., 2023; 2020; 2019a;b) Yet, few works have investigated the capability of local learning on value estimation.

**Action Conditioning of Value Estimators:** An important design choice in value estimation is whether the network is conditioned on the action. Early neural value estimation methods Riedmiller (2005) incorporated action conditioning by incorporating both state and action as model inputs. With the advent of deep neural network approaches such as DQN, practices began to diverge. Purely value-based methods like DQN are typically only state-conditioned, with action-specific predictions produced at the output layer by indexing over action values. This design is computationally efficient and well-suited for discrete tasks with low-dimensional action spaces. In contrast, actor–critic methods developed for high-dimensional continuous control tasks Lillicrap et al. (2015); Haarnoja et al. (2018a) condition on both state and action at the input of their critic networks. Although this distinction is largely arbitrary in backpropagation-based architectures and can be adapted to the task, we show that action conditioning at model inputs is strictly preferable for local RL.

**Local and Decentralized Reinforcement Learning:** The concept of decentralized RL can be dated back to the dawn of RL. Klopf (1982) introduced the idea of the hedonistic neuron, which hypothesized that each of our neurons may be guided by their independent rewards. Instead of being a miniscule part of a large operating neural system, each neuron may be an RL agent itself. In modern RL literature, the localized formulation of RL methods can be related to the multi-agent RL (MARL) setup, where multiple independent agents can be designed to cooperate well toward maximizing their joint rewards (Tan, 1993; Foerster et al., 2017; Palmer et al., 2017; Su et al., 2022; Lauer & Riedmiller, 2000; Jiang & Lu, 2023; De Witt et al., 2020; Su & Lu, 2022; Su & Lu; Arslan & Yüksel, 2016; Jin et al., 2021). Conveniently, we can frame the problem of training RL using local objectives as a MARL problem where each agent represents different modules within a network. Recently, Seyde et al. (2022) has explored a similar approach for the continuous control problem, showing that using a separate critics network for each fixed action after action discretization works surprisingly well. Guan et al. (2024) builds upon the FF architecture, showing that a network with nonlinear local operations, decentralized objectives, and top-down connections across the temporal dimension can exceed state-of-the-art methods trained end-to-end. We extend upon this literature of decentralized methods for value estimation.

## 3 BACKGROUND

**Forward-Forward (FF):** The FF Algorithm (Hinton, 2022), as its name denotes, uses two forward passes instead of one forward pass and one backward pass used in backpropagation. The first forward pass carries the positive data, or real data, while the second pass carries the negative data, or fake data either manually defined or synthetically generated by the network. The network is then trained by maximizing the goodness of each layer in the positive pass, while minimizing the goodness of each layer in the negative pass. The definition of goodness based on a hidden vector $z$ is as follows:

$$G_z = \sum_{z_i \in z} z_i^2. \tag{1}$$

In layman's terms, this equation represents the sum of squares of all activations over $L$, a measure of the magnitude and orientation of the activation vector. By training its layers greedily, FF is biologically plausible and could serve as a model for our future discovery of the inner mechanisms of the human brain.

**Value Estimation in Deep RL:** Estimation of the value function is core to RL. In layman's terms, the value function measures the expected sum of future rewards after discounting given a current state. A similar formulation can be constructed when we are interested in the goodness of a state-action pair, which is usually termed the Q-Function. Formally,

$$Q_\pi(s, a) = \mathbb{E}_\pi \left[ \sum_{k=0}^{\infty} \gamma^k R_{t+k+1} \middle| S_t = s, A_t = a \right]. \tag{2}$$

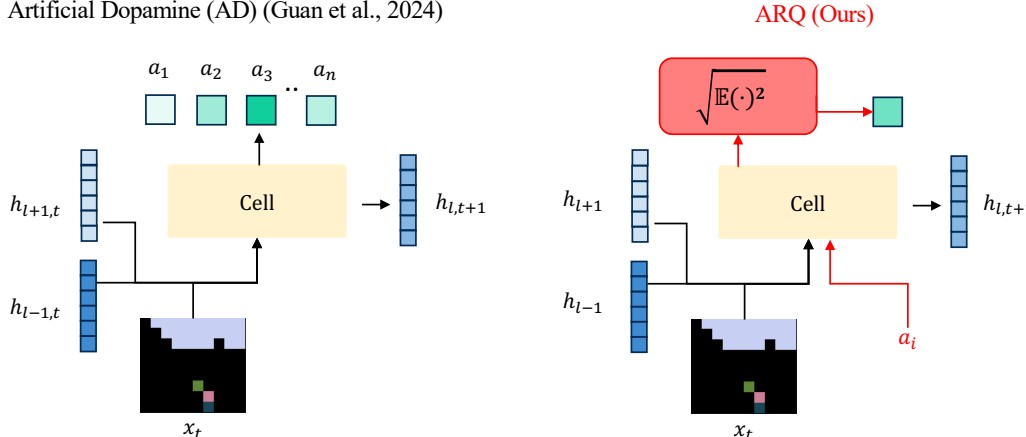

Figure 2: High-level computation diagram between Guan et al. (2024) and ARQ. Key implementations of ARQ are highlighted in red. AD cells take activations (highlighted in blue, darker color means earlier layer) and the state observation as input and produces a vector of size $n_a$, each indicating the value prediction of an action candidate. Our ARQ takes activations, the state observation, and the action candidate as input, and produces a hidden vector of arbitrary size, before passing it through a root mean squared function to yield a scalar prediction.

A widely used class of methods for value estimation is temporal difference (TD) learning (Sutton, 1988), which bootstraps value estimates by blending immediate rewards with future predictions, allowing for online, incremental updates. This method paved the way for the development of many subsequent approaches, particularly Q-learning. Take a Q-Function $Q(s, a)$. To update the function given an experience $(S_t, a, r, S_{t+1})$, Q-learning makes the following iterative update

$$Q^{(i+1)}(S_t, A_t) = Q^{(i)}(S_t, A_t) + \alpha(R_t + \gamma \max_{a'} Q^{(i)}(S_{t+1}, a') - Q^{(i)}(S_t, A_t)), \qquad (3)$$

where $\gamma$ is a discounting factor, $\alpha$ is a pre-determined learning rate, and $a'$ represents any possible actions in the next step.

Recently, the rise of neural networks pushed q-learning to new heights. Mnih et al. (2013) proposed DQN, approximating Q-values using a deep neural network. Based on the Bellman equation, DQN constructs a mean squared error function as the objective, namely

$$L_\theta = \left( R_t + \gamma \max_{a'} Q_\theta(S_{t+1}, a') - Q_\theta(S_t, A_t) \right)^2. \qquad (4)$$

Mnih et al. (2013) tested their agents on the Atari 2600 environment, and show that a convolutional neural network trained in this fashion is able to achieve near-human performance level from raw pixel inputs, a feat previously considered far-fetched.

**Artificial Dopamine (AD):** AD (Guan et al., 2024) trains a local RL agent using Q-learning. An AD network is consisted of multiple AD cells, each of which makes an independent estimation of $Q(S_t, A_t)$. To yield a scalar estimation, each AD cell adopts an attention-like mechanism to compute a weighted sum of its hidden activations using weights from a separate linear projection, effectively incorporating nonlinearity while maintaining backprop-free. Additionally, each AD cell takes inputs from the layer below, the layer above, and also the raw state observation, enabling skip connections, top-down connections, and information flow throughout the temporal dimension in an RL environment. Mathematically, an AD cell at depth $l$ conducts the following operations,

$$X = \text{concat}(s_t, h_t^{l-1}, h_{t-1}^{l+1}), \qquad (5)$$

$$h_t^l = \text{ReLU}(W_h X), \qquad (6)$$

$$Q(s_t, a_t) = \tanh(X^T W_{att_2}^T W_{att_1} X) h_t^l, \qquad (7)$$

where $h_t^l$ represents the activation of the AD cell at time $t$ and depth $l$. While this attention-like mechanism brings exciting nonlinearity to a single AD cell without the need for backpropagation, the scalar nature of $Q(s_t, a_t)$ implies that the dimensionality of $W_{att}$ must be limited by the size of the action space. We aim to remove this constraint.

## 4 ARQ: ACTION-CONDITIONED ROOT MEAN SQUARED Q-FUNCTION

In the context of FF, the goodness function measures the likelihood of the observation to come from the postive distribution. In the context of RL, the concept of value measures the expected sum of future rewards for the trajectories starting from a given state. We observe a connection—both denote a measure of the current input's desirability to an agent. Could the association between goodness and value be exploited to unleash the capacity of local RL networks? In this section, we introduce a novel vector-based training mechanism for local value estimation that can be used out-of-the-box. We term it the Action-conditioned Root mean squared Q-Function (ARQ).

### 4.1 ARQ

Take a state $s$ and an action $a$. Based on the Bellman equation, we are interested in finding

$$Q_*(s,a) = \mathbb{E}_\pi \left[ R_t + \gamma \max_{a'} Q_*(S_{t+1}, a') \big| S_t = s, A_t = a \right]. \tag{8}$$

Inspired by the association between the concept of goodness from FF and the concept of value in RL, we directly approximate $Q(s,a)$ using the goodness function. Given a hidden vector $z$, which can be either an intermediate action or an output embedding from a neural network. Instead of taking the sum of each vector unit squared, we make a small modification and take the root mean squared (RMS) function of the vector after mean subtraction to prevent its goodness values from exploding as we scale up the number of units. This is equivalent to the standard deviation of the hidden vector. In mathematical terms, we compute the estimated value of applying action $a$ on state $s$ using

$$\mu_y = \mathbb{E}_{y_i \in y} y_i, \quad Q_\theta(s,a) = \sqrt{\mathbb{E}_{y_i \in y} (y_i - \mu_y)^2}, \tag{9}$$

where $\theta$ denotes the parameters of the network and $z$ denotes a hidden vector produced by the network.

To train this network, we update our weights using the same mean squared objective function as previous Q-learning methods (Mnih et al., 2013). Namely,

$$L_\theta = \left( R_t + \gamma \max_{a'} Q_\theta(S_{t+1}, a') - Q_\theta(S_t, A_t) \right)^2. \tag{10}$$

Note that it is possible to sample positive and negative data in order to train in the same contrastive fashion as the original FF algorithm, particularly when our method is used with a training mechanism that maintains a replay buffer. We leave this for future investigations to keep our method versatile.

ARQ can be implemented out-of-the-box in place of the standard Q-learning formulation. Given any intermediate vector produced by an arbitrary neural network architecture, ARQ can extract scalar statistics that serve as a prediction for the estimated value without any parameters. This property allows architectures designed for local RL to enjoy greater flexibility.

**Action Conditioning:** Due to the nature of goodness functions to produce scalar values, it is natural to implement ARQ with action conditioning at the model input. Concretely, to estimate $Q_\theta(s,a)$, the neural network $\theta$ takes both the state vector $s$ and the action vector $a$ as inputs and outputs a single scalar prediction. This contrasts with implementations such as Mnih et al. (2013) and Guan et al. (2024), where the model receives only the state vector $s$ and produces an output of dimension $n_a$, with each entry corresponding to the value of a discrete action. We demonstrate in Section 5 that this minor design decision is critical to the performance of local RL methods. For tasks with discrete action spaces, we use a one-hot vector to represent an action candidate. For tasks with continuous action spaces, we apply bang-bang discretization on the action space following Seyde et al. (2021) and condition the network on the binary action vector.

### 4.2 IMPLEMENTATION

To evaluate our method against state-of-the-art local RL architectures, we implement AR on top of Guan et al. (2024).

Our implementation is consisted of multiple cells stacked together, each of which takes inputs from the layer below, the layer above, the input observation, and an action candidate $a_t$ to make an estimation of $Q(s_t, a_t)$. Each cell adopts a similar attention-like mechanism as Guan et al. (2024). After the attention mechanism, we apply the goodness function on the intermediate vector after the attention computation. Specifically, a cell at depth $l$ conducts the following operations,

$$X = \text{concat}(s_t, h_t^{l-1}, h_{t-1}^{l+1}, a_t), \tag{11}$$

$$h_t^l = \text{ReLU}(W_h X), \tag{12}$$

$$y_t^l = \tanh(X^T W_{att_2}^T W_{att_1} X) h_t^l, \tag{13}$$

$$\mu_y = \mathop{\mathbb{E}}_{y_i \in y_t^l} y_i, \quad Q(s_t, a_t) = \sqrt{\mathop{\mathbb{E}}_{y_i \in y_t^l} (y_i - \mu_y)^2}, \tag{14}$$

Gradients are passed only within each cell to ensure the architecture is backprop-free.

Pseudocode for AD and ARQ is provided in Figure 3. In both architectures, the intermediate quantities $Z_1$, $Z_2$, and $h_t^l$ play roles analogous to the *query*, *key*, and *value* vectors in self-attention. The computation of $Z_2^T Z_1$ (Line 8, Algorithm 2) produces a dimension-wise interaction map that determines how information is redistributed across latent dimensions, similar in spirit to an attention mechanism but applied over feature dimensions rather than token positions. The key distinction in ARQ is that $Z_2$ is not restricted to have width $n_a$; instead, its dimensionality can be chosen freely. This flexibility allows ARQ to learn richer state–action interactions than AD, whose dimensionality is constrained by the cardinality of the action space.

| **Algorithm 1** AD (Guan et al., 2024) | **Algorithm 2** ARQ (Ours) |
|---|---|
| 1: $X \leftarrow [s_t, h_t^{l-1}, h_{t-1}^{l+1}]$ | 1: $X \leftarrow [s_t, h_t^{l-1}, h_{t-1}^{l+1}]$ |
| 2: $h_t^l \leftarrow \text{LayerNorm}(\text{ReLU}(W_h X))$ | 2: $h_t^l \leftarrow \text{LayerNorm}(\text{ReLU}(W_h X))$ |
| 3: $\qquad\qquad\qquad\qquad\quad \triangleright$ Dimension: $d$ | 3: $\qquad\qquad\qquad\qquad\quad \triangleright$ Dimension: $d$ |
| 4: $Z_1 \leftarrow W_{att_1} X \quad \triangleright$ Dimension: $n_a \times d$ | 4: Repeat $X$ along batch dim $n_a$ times |
| 5: $Z_2 \leftarrow W_{att_2} X \quad \triangleright$ Dimension: $d_{att} \times n_a$ | 5: $X \leftarrow [X, a_t] \qquad \triangleright$ Action conditioning |
| 6: $W \leftarrow Z_2^\top Z_1 \qquad \triangleright$ Dimension: $n_a \times d$ | 6: $Z_1 \leftarrow W_{att_1} X \quad \triangleright$ Dimension: $d_{att} \times d$ |
| 7: $W \leftarrow \text{LayerNorm}(\tanh(W))$ | 7: $Z_2 \leftarrow W_{att_2} X \quad \triangleright$ Dimension: $d_{att} \times d$ |
| 8: $Q \leftarrow W h_t^l \qquad\quad \triangleright$ Dimension: $n_a$ | 8: $W \leftarrow Z_2^\top Z_1 \qquad \triangleright$ Dimension: $d \times d$ |
| | 9: $W \leftarrow \text{LayerNorm}(\tanh(W))$ |
| | 10: $y \leftarrow W h_t^l \qquad\qquad \triangleright$ Dimension: $d$ |
| | 11: $Q \leftarrow \text{RMSQ}(y)$ |

Figure 3: Comparison of AD and ARQ implemented on top of AD. For ARQ, action conditioning is applied as part of the input (Line 5,6, Algorithm 2). Note that ARQ allows $Z_2$ and $y$ to have dimension $d$ (red), which can be arbitrary, while AD fixes it at $n_a$ (blue), one for each action output.

**Why ARQ benefits local Q-learning?** As demonstrated in Figure 3, ARQ allows the hidden output to have arbitrary dimensions. We conjecture that ARQ's flexibility to account for arbitrary hidden dimensions allows it to take full advantage of non-linearity within each AD cell. Furthermore, ARQ applies action conditioning at the model input, rather than using vector indices at the output layer as conditioning. We conjecture that this allows the entire module to produce representation specific to each state-action pair, rather than action-agnostic information based on only the observation. Combining these two properties, ARQ exploits the full capacity of the attention-like mechanism that modern local RL methods operates on, allowing greater expressivity of each state-action pair.

## 5 EXPERIMENTS

**Benchmarks:** We test ARQ on the MinAtar benchmark (Young & Tian, 2019) and the DeepMind Control (DMC) Suite (Tassa et al., 2018) following Guan et al. (2024). MinAtar is a miniaturized version of the Atari 2600 games, using 10x10 grids instead of 210x160 frames as inputs. The DMC Suite is a benchmark for continuous control tasks featuring low-level observations and actions, designed to evaluate the performance of RL methods in physics-based environments. Both benchmarks involve low-dimensional inputs and outputs instead of high-dimensional raw sensory inputs, making them appropriate testbeds for evaluating the decision-making ability of local methods in simple environments.

**Baselines:** For comparisons with cutting-edge local RL methods, we compare our results with AD for both benchmarks. To evaluate our methods against backprop-based algorithms, we also compare our method against DQN for MinAtar. DQN is a widely used baseline that trains deep neural

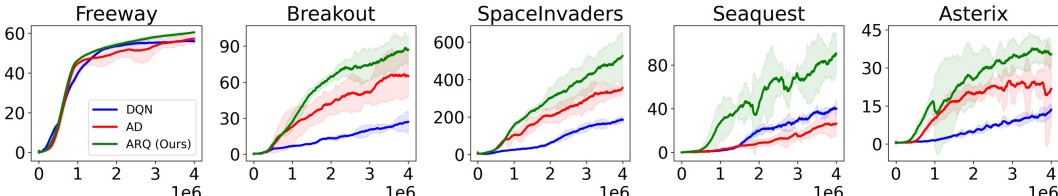

Figure 4: Training performance on the MinAtar games, compared between DQN (blue), AD (red), and ARQ (green). The x-axis denotes the number of training steps (in millions), and the y-axis indicates average episodic returns. Shaded regions represent 95% confidence intervals across 5 seeds. We find that ARQ consistently outperforms AD in all MinAtar games, while outperforming DQN in some games.

networks to directly compute scalar Q-values through backpropagation. We follow the DQN implementation used by Guan et al. (2024).

**Implementation Details:** Following Guan et al. (2024), we use a three-layer fully-connected network, with hidden dimensions being 400, 200, and 200 for MinAtar. We use a three-layer network with hidden dimensions 128, 96, and 96 for DMC tasks. We use a replay buffer and a target network for stability. We incorporate skip connections from the input and top-down connections from the layer above. For all experiments, we use an epsilon-greedy policy with linear decay from 1 to 0.01 using an exploration fraction of 0.1. We run our experiments for 4 million steps, where the model starts learning from step 50,000. Learning rate is set fixed at 1e-4. A batch size of 512 is used. For MinAtar, we condition on action candidates by passing them as one-hot vectors into the network. For DMC tasks, we discretize our action space and condition action vectors as model inputs.

**Main Results:** As presented in Figure 4, we run each experiment with five different random seeds and plot their average returns over 100-episode windows along with their 95% confidence intervals in shadows. We also calculated the average returns of the last 100 episodes of each training run to obtain a quantitative measure of the final performance of our method, which can be found in Table 1. As demonstrated, ARQ consistently outperforms AD in all MinAtar games. Surprisingly, ARQ also outperforms DQN in all games. In DMC Suite tasks, ARQ achieves superior returns compared to AD, while also exceeding back-prop based methods in most games. A possible explanation is that ARQ benefits from localized TD updates, reduced gradient path length, and the variance reduction effect of layerwise averaging, which together can lead to more stable and efficient learning than fully backpropagated networks.

**Game Analysis:** We note that ARQ outperforms DQN by a wide margin on Breakout and SpaceInvaders. Both of these games operate on similar mechanisms: players aim to remove targets by controlling projectile interactions of objects. To yield higher scores, players need to perform combos of actions to yield higher scores, for instance moving to a sweet spot then waiting for the target to arrive before firing a bullet. We argue that top-down connections in AD provide temporal coherence, which allows our agents to perform sequences of actions smoothly. Additionally, we note that while AD fails to match DQN on Seaquest, ARQ surpasses DQN. Seaquest is a game involving firing bullets to remove enemies, with an additional rule that players need to manage an oxygen tank by surfacing above water to refill their tank. This represents that the policy distribution can be bi-modal such that attacking enemies and refilling tanks are both locally optimal policies. We hypothesize that by applying action conditioning, ARQ can capture these policy structures more effectively than AD, which is only state-conditioned.

**Effect of Action Conditioning at Input:** How does action conditioning affect the performance of local RL methods? To investigate, we conduct ablation experiments on two games from MinAtar, Breakout and SpaceInvaders, using both AD and ARQ. The results can be found in Figure 5. We find a significant improvement when actions are conditioned at the input instead of at the output. To further understand this difference, we analyzed the hidden activations using PCA and compared them against predicted Q-values as presented in Figure 6. Without action conditioning, activations cluster almost entirely by action identity and show no meaningful correlation with Q-values, indicating that action-related variance dominates the representation space. With action conditioning, representations become more state-driven and exhibit a mild positive relationship with Q-values, suggesting that the model can allocate capacity toward value-relevant structure rather than implicitly inferring

Table 1: Performance of previous methods and ARQ on MinAtar and DeepMind Control (DMC) tasks. Reported as mean ± 95% confidence intervals across 5 random seeds.

| **MinAtar** | Freeway | Breakout | SpaceInvaders | Seaquest | Asterix |
|---|---|---|---|---|---|
| w/ back-prop | | | | | |
| DQN | $55.86 \pm 0.64$ | $27.09 \pm 11.48$ | $188.03 \pm 31.62$ | $37.96 \pm 18.56$ | $13.60 \pm 2.16$ |
| **w/o back-prop** | | | | | |
| AD | $57.12 \pm 2.70$ | $63.76 \pm 17.70$ | $363.49 \pm 36.92$ | $27.83 \pm 10.97$ | $22.01 \pm 10.26$ |
| ARQ (Ours) | $\mathbf{60.74 \pm 0.54}$ | $\mathbf{87.84 \pm 11.10}$ | $\mathbf{544.99 \pm 88.10}$ | $\mathbf{96.45 \pm 21.44}$ | $\mathbf{35.32 \pm 5.33}$ |
| **DMC** | Walker Walk | Walker Run | Hopper Hop | Cheetah Run | Reacher Hard |
| w/ back-prop | | | | | |
| TD-MPC2 | $958.80 \pm 2.58$ | $834.07 \pm 20.26$ | $348.55 \pm 53.30$ | $808.46 \pm 184.20$ | $934.84 \pm 13.72$ |
| SAC | $980.43 \pm 3.26$ | $895.02 \pm 92.70$ | $319.46 \pm 62.42$ | $917.40 \pm 4.90$ | $980.01 \pm 2.38$ |
| **w/o back-prop** | | | | | |
| AD | $975.30 \pm 2.10$ | $762.51 \pm 4.86$ | $470.95 \pm 78.14$ | $831.57 \pm 31.80$ | $955.93 \pm 18.36$ |
| ARQ (Ours) | $\mathbf{976.33 \pm 1.04}$ | $\mathbf{771.15 \pm 5.08}$ | $\mathbf{516.23 \pm 34.72}$ | $\mathbf{880.61 \pm 24.04}$ | $\mathbf{973.66 \pm 9.98}$ |

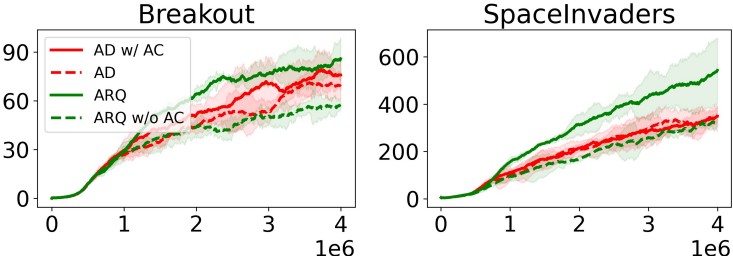

Figure 5: Ablation on action conditioning for AD and ARQ. Action conditioning substantially improves performance. Shaded regions represent 95% confidence intervals. Note that this improvement is particularly significant for ARQ, with average returns of ~85 vs. ~55, a 50% improvement. This indicates that the combination of the RMS function and action conditioning makes ARQ effective.

action identity. Interestingly, this design choice provides only a slight improvement for AD, while yielding a significant increase in performance for ARQ. We conjecture this is due to the increase in the capacity of each cell to capture the granularity within each specific state–action pair, while AD saturates with action-agnostic information.

**Effect of Goodness Nonlinearities:** One question that naturally arises is the choice of the goodness function. Does the RMS function perform superiorly compared to other functions? We ablate on this design choice and conduct experiments on two games from MinAtar, Breakout and SpaceInvaders. As shown in Table 2, we find that using the RMS goodness functions yields superior performance, followed by the mean and the mean squared function. Beyond performance, our analysis suggests that RMS maintains healthier activation magnitudes throughout training, whereas using mean squared function produces extremely large early goodness values that later suppress activation norms (see Figure 7). This stabiliza-

Table 2: Our method Using Different Nonlinearities Compared in MinAtar Breakout. 'MS' is short for the mean squared function and 'Var' is short for variance. Default ARQ uses the root mean squared (RMS) function. Reported as mean ± 95% confidence intervals.

| Nonlinearity | Breakout | SpaceInvaders |
|---|---|---|
| Ours-ARQ | $\mathbf{87.84 \pm 11.10}$ | $\mathbf{544.99 \pm 88.10}$ |
| Ours-Mean | $79.84 \pm 26.46$ | $500.13 \pm 95.56$ |
| Ours-MS | $82.10 \pm 6.56$ | $434.88 \pm 28.74$ |
| Ours-Var | $81.34 \pm 0.78$ | $416.46 \pm 133.2$ |
| AD | $67.40 \pm 8.02$ | $369.96 \pm 46.92$ |

tion effect likely preserves a richer and more expressive representation space, contributing to RMS's empirical advantage. However, we note that all functions perform superiorly compared with AD, which demonstrates the versatility of our method.

**Is it because ARQ has more hidden units?** Compared with AD, ARQ employs a larger number of parameters since ARQ allows an arbitrary dimension for its hidden vectors. Could ARQ, however,

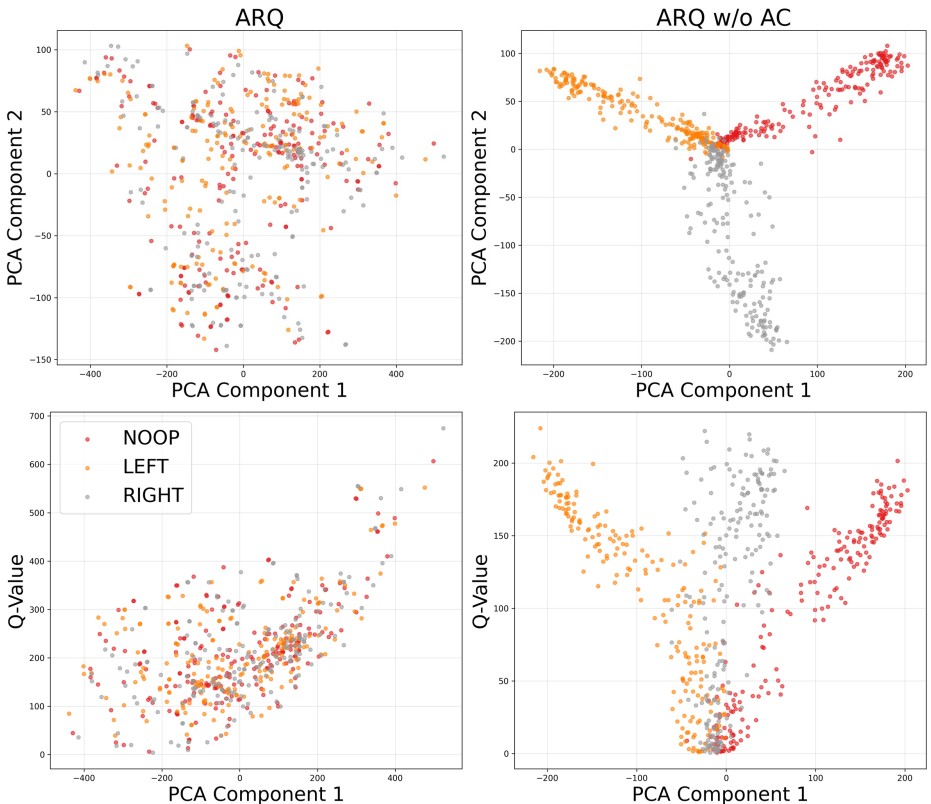

Figure 6: **Representation analysis of ARQ with and without action conditioning.** We train two agents—ARQ with and without action conditioning (AC)—on MinAtar Breakout, then randomly sample 200 states from each trained policy. For each state–action pair, we extract the hidden activations from Layer 0 and visualize them using 2-component PCA. The top row shows how mPCA components cluster across different actions. Without AC, activations form tight clusters determined almost entirely by action identity, indicating that the agent must implicitly encode action information inside the representation. With AC, the activations are more entangled and state-driven. The bottom row plots the first PCA component against the predicted Q-values. With AC, ARQ exhibits a mild positive correlation between latent structure and Q-values, while the non-AC model shows no meaningful correlation and remains dominated by action-specific clustering.

simply achieve the same improvement with mere scaling? We conduct experiments on AD and ARQ with the same number of total parameters to answer this question. Across different ratios of total parameters (compared with the original AD as a baseline), we run both AD and ARQ on the MinAtar Breakout game with two different random seeds. As shown in Table 3, ARQ consistently outperforms AD across all scales. This verifies the effectiveness of our method beyond scale.

**Neurons Are Sensitive to Different Scenarios:** How does our method learn through a goodness function? We investigate its inner mechanism by visualizing the activations at each layer under different states. As illustrated in Figure 8, we find that the hidden activations tend to show larger magnitudes under "correct" state-action pairs. For instance, in scenarios where the agent should move right to accurately catch the incoming ball, neurons in the hidden activations show the largest magnitude when the action input matches correspondingly. In-

Table 3: AD vs. ARQ Across Multiple Scales for MinAtar Breakout. Reported as mean ± 95% confidence intervals.

| Scale Ratio | AD | ARQ |
|---|---|---|
| 0.5× | 66.34 ± 5.15 | **68.12 ± 5.65** |
| 1× | 64.20 ± 1.90 | **86.26 ± 0.66** |
| 1.5× | 56.63 ± 5.39 | **70.40 ± 3.98** |
| 2× | 59.79 ± 4.77 | **83.26 ± 2.32** |

terestingly, we observe that different neurons are, in general, activated to different degrees for various action candidates. This implies our objective function could be encouraging specialized neurons, each of which is responsible for recognizing certain categories of positive signals.

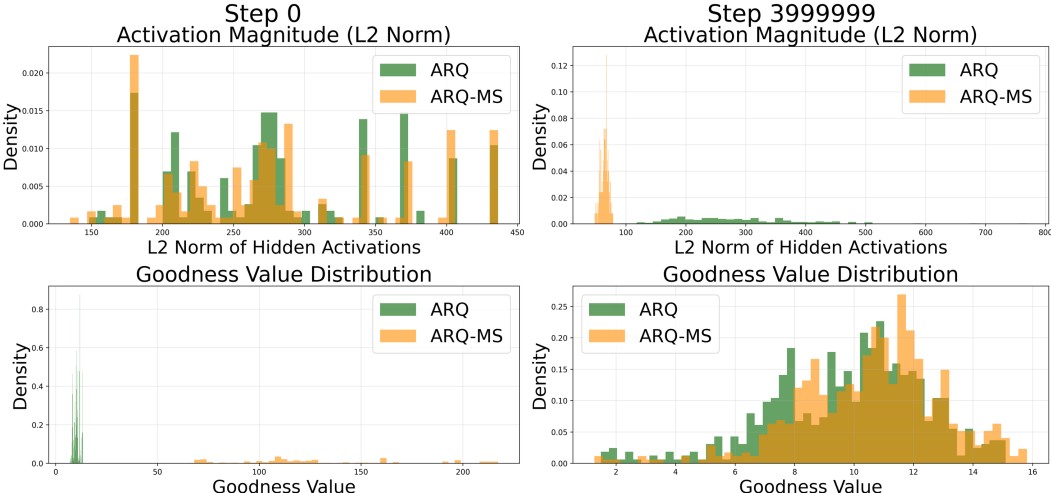

Figure 7: Analysis of ARQ (RMS goodness) vs. ARQ-MS (mean-squared goodness) on MinAtar–Breakout. We evaluate hidden activations and goodness values of both agents over 200 randomly sampled states at both the start and end of training. **Top row:** Distribution of L2 norms of hidden activations. Early in training, ARQ-MS exhibits extremely large goodness values and compressed activation magnitudes, while ARQ maintains broader, more expressive activation scales. By the end of training, ARQ continues to support significantly larger activation norms, whereas ARQ-MS activations remain narrowly concentrated. **Bottom row:** Distribution of scalar goodness values derived from each hidden vector. ARQ produces moderate, stable goodness magnitudes throughout training, while ARQ-MS shows large initial spikes followed by sharply reduced variability.

## 6 DISCUSSION

Previous studies on biologically plausible learning have largely focused on the search for a biologically plausible mechanism for performing gradient updates. As we approach the era of experience, we argue that a biologically plausible paradigm for learning can be equally meaningful to guide us towards the mystery behind how biological brains learn. Reward-centric environments provide a biologically grounded paradigm, aligning with the evolutionary role of survival signals and behavioral shaping through positive or negative reinforcement. The structure of such environments mirrors the ecological settings in which animals adaptively refine behavior through trial-and-error interactions, suggesting that learning systems shaped by rewards may naturally emerge in both artificial and biological agents. Additionally, temporal difference methods are an ideal candidate. It has been shown that biological neurons learn through temporal difference, with hormones conveying the prediction error as a source of learning signal to independent neurons (Schultz et al., 1997b; Bayer & Glimcher, 2005). On the other hand, reinforcement learning has largely focused on learning through interactions with an agent's surrounding environment, and maximizing its rewards through centralized value estimation. Yet, increasing neuroscientific evidence has shown that neurons make decentralized, independent value estimations (Tsutsui et al., 2016; Knutson et al., 2005). Few work in the RL community has investigated whether this biological phenomenon has practical implications. ARQ is an effort towards this direction as each cell in our network can be seen as a decentralized value estimator.

## 7 CONCLUSION

This work proposes Action-conditioned Root mean squared Q-Function (ARQ), a vector-based alternative to scalar Q-learning for backprop-free local learning. ARQ enables arbitrary hidden dimensions and improved expressivity by extracting value predictions from hidden activations and applying action conditioning at the model input. We show that, when applied on RL environments, ARQ performs superiorly compared to current local methods, while also outperforming backprop-based methods on some games. Whereas current biologically plausible algorithms are mostly based on the supervised setting, our study suggests that exploring local learning within reinforcement learning may provide a promising avenue for future research in both domains.

## LLM USAGE STATEMENT

LLMs were used in this work to refine certain textual phrasing and to generate minor code elements related to visualization and checkpointing.

## REPRODUCIBILITY STATEMENT

We provide training code for all experiments as supplementary material. All hyperparameters, random seeds, and implementation details are described in Section 5. Our experiments were run on single NVIDIA A4000 GPUs or NVIDIA L40S GPUs with training times ranging from 8-72 hours depending on the task. We used only publicly available benchmarks (MinAtar and DeepMind Control Suite). Together, these resources should enable full reproduction of our results.

## ACKNOWLEDGEMENT

We thank Jonas Guan for his help in reproducing AD. MR is supported by Visko AI and the Institute of Information & Communications Technology Planning Evaluation (IITP) under grant RS-2024-00469482, funded by the Ministry of Science and ICT (MSIT) of the Republic of Korea in connection with the Global AI Frontier Lab International Collaborative Research. The compute is supported by the NYU High Performance Computing resources, services, and staff expertise.

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

## A Visualization of ARQ Activations

Figure 8 provides the full activation visualizations referenced in the main text, illustrating how neuron responses vary across different state–action scenarios in Breakout.

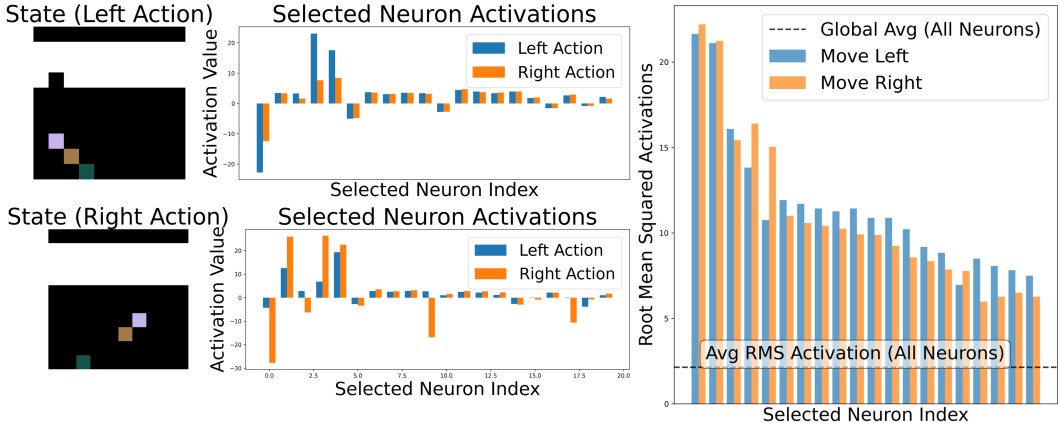

Figure 8: Visualization of Neurons in Layer 0 under Different Scenarios in Breakout Game. 20 neurons w/ highest average activities are visualized. **Top Left:** When the ball is approaching towards the left side of the brick, neurons show larger magnitude when the action candidate is to "move left", prompting the agent to move towards the ball. **Bottom Left:** When the ball approaches the right side of the brick, neurons show larger magnitude when the action candidate is to "move right". **Right:** The average root mean squared (RMS) activations of 20 top neurons across 100 states is collected. Note that top neurons exhibit significantly larger RMS activations than the average RMS activation, implying that these neurons are "dominant" neurons. While most neurons demonstrate similar magnitude between both actions, some neurons appear to be more specialized.

## B Hyperparameters

We summarize all hyperparameters used for MinAtar and DeepMind Control Suite experiments in Table 4, consolidating the network architectures, training settings, and optimization details for full reproducibility.

| Hyperparameter | MinAtar | DMC Suite |
|---|---|---|
| Network Architecture | 3-layer MLP | 3-layer MLP |
| Hidden Dimensions | 400-200-200 | 128-96-96 |
| Optimizer | Adam | Adam |
| Discount Factor ($\gamma$) | 0.99 | 0.99 |
| Learning Rate | $1 \times 10^{-4}$ | $1 \times 10^{-4}$ |
| Batch Size | 512 | 512 |
| Replay Buffer Size | 4M transitions | 4M transitions |
| Target Network | Yes | Yes |
| Exploration Strategy | $\epsilon$-greedy | $\epsilon$-greedy (on discretized actions) |
| $\epsilon$ Schedule | Linear: $1.0 \rightarrow 0.05$ | Linear: $1.0 \rightarrow 0.05$ |
| Exploration Fraction | 0.1 | 0.1 |
| Learning Starts | 50,000 steps | 50,000 steps |
| Training Steps | 4M | 4M |
| Action Representation | One-hot | Bang-bang discretization |
| Random Seeds | 5 | 5 |

Table 4: Consolidated hyperparameters for MinAtar and DeepMind Control (DMC) experiments. All settings follow the implementation described in Section 5 of the paper, with Adam optimization, a shared replay buffer of 4M transitions, and identical training schedules. Action conditioning is applied for ARQ by default.

## C    DETAILED COMPUTATIONAL DIAGRAM

In this section, we provide a detailed visualization of the computation flow in both AD and ARQ, illustrating how information propagates across layers and how each cell locally estimates Q-values.

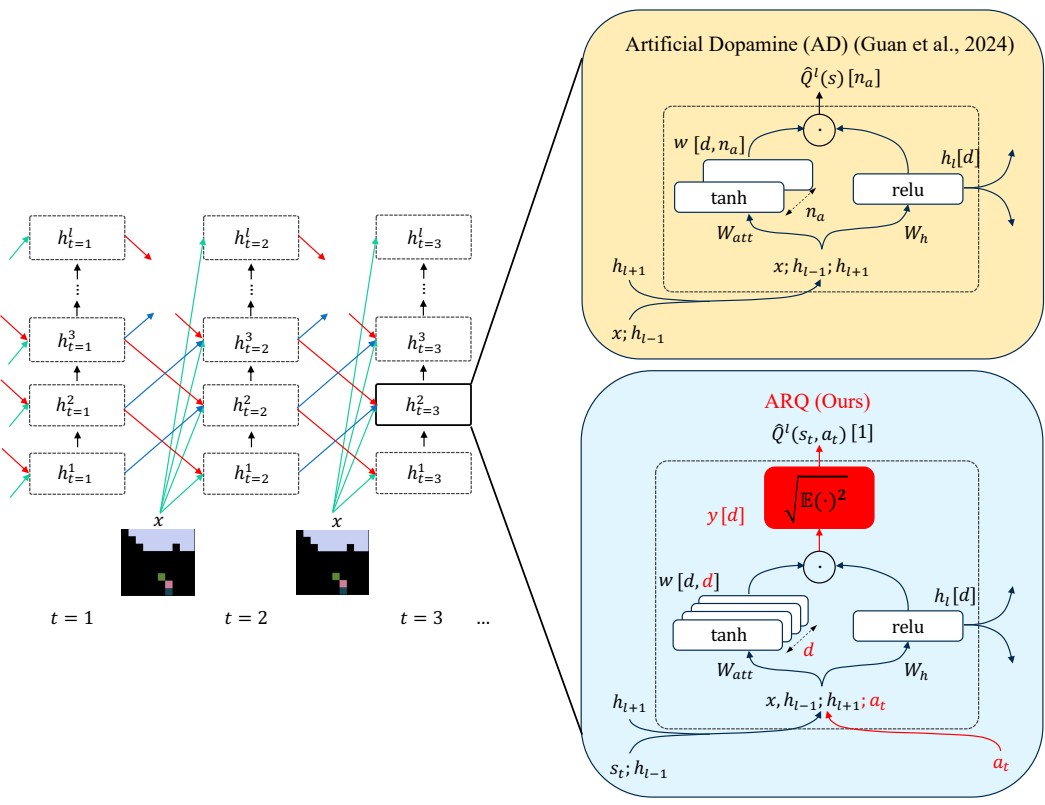

Figure 9: Detailed computation diagram. Key implementations of ARQ are highlighted in red. **Left:** Overall information processing across layers. Each cell receives input from the raw observation, the layer below, and the layer above, and its activations are passed forward to the next temporal step. **Top Right:** Cell Computation in Artificial Dopamine (AD); $n_a$ Q-value estimates are produced by dot-products between the attentional weights and the hidden states. **Bottom Right:** Our ARQ implements root mean squared functions for value estimation along with action-conditioned inputs.

## D    DEEPMIND CONTROL SUITE EXPERIMENTAL FIGURES

Figure 10 presents the full training curves for all DeepMind Control Suite environments. ARQ consistently outperforms AD across tasks and achieves performance comparable to strong backpropagation-based methods such as TD-MPC2 and SAC.

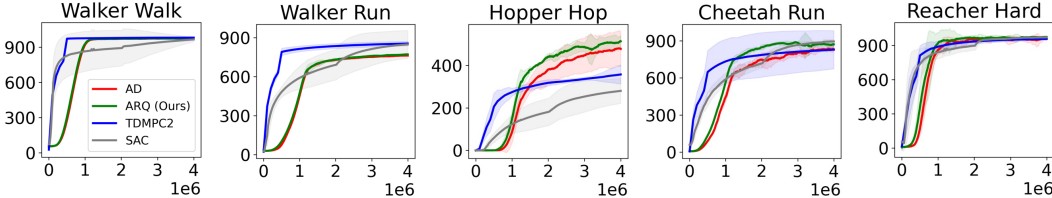

Figure 10: Training performance on the DeepMind Control Suite, compared between AD (red), ARQ (green), TD-MPC2 (blue), and SAC (gray). The x-axis shows training steps (in millions), and the y-axis denotes average episodic returns. Shaded regions indicate 95% confidence intervals across 5 seeds. Across all environments, ARQ consistently improves over AD and achieves performance competitive with backpropagation-based methods.

# E    NONLINEARITY ABLATION FIGURES

Figure 11 presents the per-environment learning curves comparing different nonlinear goodness functions used to extract scalar values from hidden vectors. The default ARQ, which uses the root mean squared function, outperforms all other choices. Notably, all variants outperform AD in both settings.

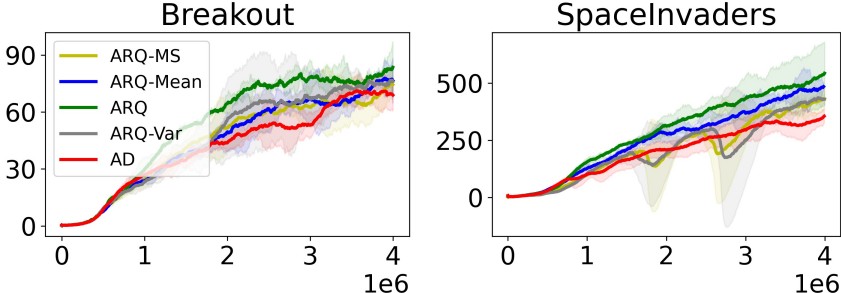

Figure 11: Comparison of different nonlinear goodness functions that may be used to collect scalar statistics from hidden vectors. 'MS' is short for the mean squared function and 'Var' is short for variance. Default ARQ uses the root mean squared (RMS) function. Shaded regions represent 95% confidence intervals.

