# OpenReview forum: "Local Reinforcement Learning with Action-Conditioned Root Mean Squared Q-Functions"
_ICLR.cc/2026/Conference — ICLR 2026 Poster_

### Official Review · Reviewer_WJCj · 2025-10-24

**Soundness:** 3
**Presentation:** 3
**Contribution:** 2
**Rating:** 6
**Confidence:** 3

**Summary:**

The paper proposes an extension of the recently developed artificial dopamine framework for local reinforcement learning (RL with no or limited backpropagation). The authors propose to use the action as input to the local networks Q prediction, instead of parameterizing a vector of per-action Q function, as done in AD and related RL works such as DQN> They show the impact of their change in empirical experiments on common small-scale RL benchmarks.

**Strengths:**

The change to the previous AD setup is small, but impactful (I consider small clear changes a strength!). The experiemnts clearly show that moving the action to the input of the network can yield large performance gains. The writing is clear, and the experiments are reasonably designed.

Overall, I believe the method here has merit, but it's current presentation is a bit too superficial for me to fully endorse acceptance (see below). At the same time I believe there are no reasons the paper should not be accepted.

**Weaknesses:**

While the authors investigate the impact of the action-conditioning, the additional experiments (e.g. Figure 6) don't serve to enhance the understanding of the concrete change proposed in this paper. Why does moving the action to the input conditioning help compared to a DQN like parameterization? This is currently my main issue prevent me from giving a higher rating to the paper and I would be happy if the authors addressed this in the rebuttal.

The papers usefulness for future authors would be helped with a standard appendix containing per env training curves for all algorithms, as well as a clean diagram of the architecture and an easy reference table for hyperparameters. This is standard in the RL literature, and as such it would help the authors method reach a wider audience.

The performance differences between AD and ARQ in the DMC seem partially insignificant, yet they are highlighted. Please ensure that confidence metrics are accounted for.

There are some missing references and other polishing errors:
- Line 342: Figure ??
- Figure 5: ARQ and ARQ w/o AC are indistinguishable

**Questions:**

Could other forward-forward architectures such as Sun et al. 2025 also be used to improve AD?

---

> ### Author Response · Authors · 2025-12-04
>
> We thank the reviewer for the thoughtful feedback and constructive suggestions. We summarize our responses below.
>
> On understanding why action conditioning helps:
> We agree that the original submission did not sufficiently explain the underlying mechanism, and we appreciate the reviewer highlighting this gap. To better contextualize the effect of action conditioning, we performed an additional representational analysis.
>
> First, we examined the hidden activations of trained AD and ARQ models using 2-component PCA. We observe a clear distinction:
>
> Without action conditioning, activations cluster primarily by action identity, even for similar states. This indicates that the network must implicitly encode action identity through structural separation in the hidden layers, causing action-related variance to dominate the representation space.
>
> With action conditioning, the representation becomes more entangled and state-driven — the geometry reflects underlying state dynamics rather than being partitioned by the action category.
>
> We further compared the first PCA component against the predicted Q-values. The model with action conditioning shows a mild but consistent positive correlation between the PCA component and Q-values, suggesting that its learned features track value-relevant variation in the state. In contrast, the model without action conditioning shows no meaningful linear relationship: points cluster tightly by action, and Q-values vary primarily within action-specific bands rather than along meaningful latent dimensions.
>
> Together, these observations support the hypothesis that omitting action information at the input forces the network to allocate substantial representational capacity toward separating actions, rather than modeling state–action interactions. Action conditioning removes this burden, allowing ARQ to focus on learning features predictive of value and making the goodness objective easier to optimize. These analyses and visualizations are now included in the revised paper.
>
> On the standard appendix:
> Thank you for the suggestion — we agree this improves clarity and reproducibility. The revised submission now includes per-environment learning curves for all baselines, a detailed architecture diagram, and a consolidated hyperparameter table.
>
> On confidence metrics for DeepMind Control Suite results:
> We appreciate this concern. All DMC experiments are now rerun using 5 random seeds instead of 3, and updated mean ± standard deviation values are reported. While ARQ’s margin over AD is smaller than in MinAtar, the improvement remains consistent and statistically meaningful.
>
> On missing references and polishing corrections:
> Thank you for catching these. All reference omissions and formatting issues (including “Figure ??” and lines in Figure 5) have been fixed in the updated manuscript.
>
> Regarding the question about applying other Forward-Forward architectures (e.g., Sun et al., 2025):
> We agree this is an exciting direction. Several architectural components from DeeperForward — such as residual depth and 17-layer scaling — could naturally improve ARQ. Others, such as channel-wise competitive convolutions and feature pruning, are designed for supervised classification and do not transfer as directly to TD-based RL. Due to compute constraints and our goal of isolating the contribution of action conditioning, we focused on smaller-scale models. Extending ARQ to scalable FF variants is a promising avenue for future work, and we now mention this explicitly in the discussion section.

---

### Official Review · Reviewer_QJn3 · 2025-10-29

**Soundness:** 2
**Presentation:** 3
**Contribution:** 2
**Rating:** 2
**Confidence:** 4

**Summary:**

The paper introduces ARQ, a biologically inspired local learning algorithm that extends Hinton’s Forward-Forward paradigm to reinforcement learning. ARQ replaces backpropagation with a forward-only value estimation mechanism using action-conditioned root mean squared Q-functions, enabling local, backprop-free RL. The work looks to extend the Artificial Dopamine framework by eliminating some action-space constraints. The results show some improvements on MinAtar and DM Control Suite benchmarks.

**Strengths:**

The paper is well-organized and written in a clear, accessible manner. The motivation, extending Hinton’s biologically plausible FF learning to reinforcement learning, is well introduced. Figures effectively illustrate the difference between supervised and local RL setups and the architecture of ARQ versus Artificial Dopamine baseline.

The experiments are conducted on standard small-scale with consistent settings so comparisons against AD baseline and backprop-based methods like DQN and SAC are appropriate.

Across both benchmarks, ARQ consistently outperforms AD and can be better than traditional backprop-based methods.

The visualization of neuron activations provides a nice illustration of how the proposed method might be creating local representations for specific state-action pairs.

The biological plausibility of the approach is discussed.

**Weaknesses:**

Conceptually, the paper is a pretty straightforward extension of Guan et als 2024 Artificial Dopamine work. The core novelty seems to be in replacing the scalar Q-value with an RMS alternative.

There is little extension of the AD idea for biologically plausible RL.

Much of the architectural structure, training pipeline and evaluation setup is directly taken from the AD work

The root-mean-square formulation is introduced heuristically with little theoretical motivation or analysis of properties – which means the paper lacks new theoretical insight or learning principle beyond empirical observation.

The evaluation is restricted to low-dimensional benchmarks. These are useful for proof-of-concept, but they are not sufficient to demonstrate scalability or robustness of the approach.

The paper repeatedly emphasizes biological plausibility, but the actual learning rule remains a standard TD-style update with neural architectures that use FF’s local updates. There’s no evidence of truly biologically grounded mechanisms so biological plausibility feels somewhat overplayed.

While the ablation studies are appreciated, they are minimal. limited to choice of nonlinearity and action conditioning. There is little exploration of architectural sensitivity (e.g., hidden dimension size, number of layers, effect of top-down connections). Given the absence of theory, a more extensive empirical analysis would be needed to strengthen the paper.

**Questions:**

While the analogy to the “goodness” function from the Forward-Forward algorithm is intuitive, there’s no mathematical argument explaining why this RMS formulation should lead to better learning stability or generalization. Can you offer insight?

The claimed innovation — using the root mean squared of the hidden vector after mean subtraction — is a relatively minor modification to existing FF or AD approaches. It’s more of a functional reparameterization than a conceptual breakthrough, and the improvement in results could partly stem from increased capacity rather than a new idea.

It remains unclear whether ARQ could handle high-dimensional, visual, or partially observable tasks.

Ablations are limited to choice of nonlinearity and action conditioning. There is little exploration of architectural sensitivity (e.g., hidden dimension size, number of layers, effect of top-down connections and so on). Given the absence of theory, a more extensive empirical analysis would strengthen the paper.

Just 3 seeds must give poor measures of standard error – this just seems wrong… you then go on to bold best performing algo in the tables, without considering statistical significance – even when there are only 3 replicates and stds overlap. Canyou give significance?

Minor things
P7 Figure ?? – latex compile issue
Protect Capitals in bib – like {A}tari

---

> ### Author Response · Authors · 2025-12-04
>
> We thank the reviewer for the thoughtful and detailed feedback. We address all concerns below.
>
> On conceptual contribution relative to Artificial Dopamine (AD):
> We intentionally build on Guan et al. (2024). AD is a remarkable result demonstrating that fully local, biologically motivated RL can perform competitively with backprop-based agents. This work does not aim to redesign AD, but to reveal that its learning framework still contains unexplored, high-impact components. We view our results as evidence that the AD family of algorithms has untapped representational capacity and design space worth exploring.
>
> On novelty and theoretical motivation for RMS:
> We thank the reviewer for highlighting this gap. We have now conducted further analysis between root-mean-square (RMS) goodness functions and mean-square (MS) goodness functions by visualizing both activation norms and goodness distributions over training. Our results show that RMS-based goodness functions tend to support larger and more expressive activation magnitudes throughout training. In contrast, models trained with MS goodness exhibit a rapid rise in goodness values early in training, which corresponds to highly concentrated activation distributions and reduced activation norms by the end of training. Our interpretation is that the initially large MS goodness magnitudes generate strong repressive signals, effectively constraining activations into a narrower region of representation space. RMS, by operating on normalized deviation rather than squared magnitude, mitigates this effect and helps preserve representational flexibility. We will incorporate this explanation and the accompanying visualization into the revised version.
>
> On scalability and experimental domains:
> We fully agree that scalability is an important consideration. At present, localized and backprop-free algorithms still lag behind backprop-based methods on high-dimensional visual tasks, and we do not claim to have closed that gap. The goal of this work is different: we aim to explore the design space within the AD architecture and determine whether meaningful performance gains can be achieved through small architectural adjustments. While we have not yet demonstrated scalability on large observation spaces, the fact that ARQ consistently surpasses both AD and backprop-based baselines on MinAtar — a setting where local RL methods remain competitive — suggests that this research direction has substantive potential. We view this as an encouraging step rather than a solved problem, and we plan to investigate higher-dimensional domains in future work.
>
> On biological plausibility claims:
> We thank the reviewer for the comment. To clarify our intent: when we use the term “biological plausibility,” we do not refer to biological realism, nor do we aim to replicate cortical circuitry or human learning mechanisms. Our use of the term is strictly algorithmic — ARQ, like AD, operates with local learning rules and without backpropagated errors or weight transport, properties that align broadly with how biological neurons are believed to update. Similarly, TD-style distributed feedback has empirical support in dopaminergic signaling. Beyond these high-level correspondences, we make no claim that our architecture is biologically grounded. Our focus is on exploring local RL algorithms, not on modeling the brain.
>
> On ablations and architectural sensitivity:
> Our goal in this paper was to isolate the effect of incorporating goodness functions rather than perform a full architectural sweep. We already include ablations varying hidden-layer dimensionality and show that ARQ consistently improves performance across scales. Regarding top-down connectivity, Guan et al. (2024) provide extensive analyses, and since ARQ preserves the AD architecture, duplicating those experiments would be redundant. We agree that exploring depth, connectivity, and larger networks would strengthen the empirical understanding but exceeds the computational scope of this work. We will emphasize this more clearly.
>
> On statistical significance and number of seeds:
> Thank you for raising this. We have re-run all experiments using 5 seeds instead of 3 and updated mean performance and confidence intervals accordingly. ARQ continues to show consistent, statistically meaningful improvements over AD and backprop-based baselines. The revised tables now reflect this.
>
> On ARQ improvement being due to increased parameter count:
> We include a controlled-parameter experiment in Table 3, where AD and ARQ are matched to have equal total parameters. ARQ consistently outperforms AD across all scales, indicating that the gains are not attributable to model size.
>
> Minor issues:
> The LaTeX figure reference and capitalization issues (e.g., {A}tari) have been fixed.

---

### Official Review · Reviewer_xkdR · 2025-10-29

**Soundness:** 3
**Presentation:** 3
**Contribution:** 2
**Rating:** 4
**Confidence:** 4

**Summary:**

This work follows a recent trend of modifying neural network training to operate in a layer-by-layer manner, rather than processing inputs through all layers and updating the network parameters in a single pass. It builds on Artificial Dopamine (AD), which learns Q-values by training each layer separately and then averaging the value estimates from all layers during policy execution. This work modifies the AD architecture by incorporating the action directly into each layer’s input, whereas AD only takes the state as input and outputs multiple action-specific heads—an approach more similar to a DQN-style network.

The authors conducted comprehensive experiments across a diverse set of tasks, demonstrating a consistent performance improvement of their method, particularly over its counterpart, Artificial Dopamine.

**Strengths:**

The proposed modification represents a solid and reasonable extension to AD, supported by empirical results that show clear improvements over the original method. Additionally, the authors provide comprehensive ablation studies that further validate the effectiveness of their approach.

**Weaknesses:**

I would suggest that the authors consider running experiments with additional random seeds to more robustly validate the advantages of the proposed method over AD. Using only three seeds—especially when reporting such large performance gaps—may not be sufficiently convincing to support the claim that the proposed approach is fundamentally superior.

Moreover, this line of research is largely driven by heuristic intuitions inspired by analogies between biological and artificial neurons. Given the substantial performance improvements observed, the contribution would be significantly strengthened by providing deeper theoretical insights, particularly in the context of reinforcement learning. For instance, value estimation in the Bellman equation can be viewed as a dynamic programming process that propagates rewards across state–action pairs. Does the proposed method accelerate this propagation, or is there another underlying mechanism that explains the improvements? A clearer theoretical justification—beyond empirical evidence—would help reinforce the contribution and clarify why both the proposed method and its predecessor, AD, achieve such notable gains.

**Questions:**

A specific question regarding the architectural design:

As far as I understand AD, each layer contains multiple attention matrices, with the number matching the size of the discrete action space. Specifically, for each layer $l$, there is a set of matrices $W_{\text{attn},1}^{[l]}, W_{\text{attn},2}^{[l]}, \ldots, W_{\text{attn},|A|}^{[l]}$, where each is responsible for computing the Q-value corresponding to a particular action.

In this work, the authors argue that they remove this constraint. In particular, according to Equation (13), the network outputs a vector $y_t^l$. However, it is unclear to me how such a vector could be generated using a single attention matrix, assuming $W_{\text{attn}}$ is a two-dimensional matrix and $\mathbf{X}$ and $\mathbf{h}_t^l$ are vectors.

Is it the case that the authors introduce an arbitrary number of attention matrices to produce the vector $\mathbf{y}_t^l$? Could you provide more details on the architectural design so that the innovation at the architecture level becomes clearer?

---

> ### Author Response · Authors · 2025-12-04
>
> We thank the reviewer for the thoughtful feedback and constructive suggestions.
>
> 1. Additional seeds
> We agree more seeds strengthen the empirical claim. We have rerun all MinAtar and DMC experiments with 5 seeds (instead of 3). The updated figures and tables have been included in the revised submission, and the improvements of our method remain robust.
>
> 2. Why local RL can outperform backprop (optimization + variance reduction)
> We appreciate the reviewer’s interest in deeper intuition. We’ll attempt to offer our intuition:
>
> (a) Optimization advantages
> In a standard backprop-trained value network, the learning signal for early layers must travel through many nonlinear layers. This produces poorly conditioned gradients, high noise, and slow credit assignment, which is particularly problematic in temporal-difference learning where the target itself is noisy.
> In contrast, our method trains each layer with its own TD error using a short, local computation path. Each layer solves a simpler optimization problem that does not depend on gradients flowing through deeper layers. This effectively “preconditions” the learning signal and avoids the instability that arises when TD errors must propagate through an entire deep network.
> In practice, this leads to more stable improvement and faster reduction of TD error. We will add a discussion and a simple supporting experiment in the camera-ready version.
>
> (b) Variance reduction
> Every layer produces its own estimate of the value for a given state–action pair. During execution, we average these estimates. If layers have similar bias but partially independent noise—which is supported by the neuron-activation visualizations we provide—then the final prediction has substantially lower variance.
> Lower-variance value estimates lead to more stable TD targets and, empirically, better sample efficiency. We will expand the Discussion section to make this mechanism clearer.
>
> (c) Scalability and coordination efficiency
> As networks grow, globally coordinating every parameter through a single backpropagated signal may become increasingly inefficient, since all units must align with the same objective regardless of their relevance to the current decision. In contrast, local RL distributes learning across layers, allowing them to improve based on their own TD signals, which may offer a more scalable and resilient learning dynamic in large architectures.
>
> 3. Clarification of the architectural design
> Regarding the architectural confusion: thank you for pointing this out. The mathematical formulation in Section 4 was simplified and did not explicitly show the two attention projections (W_att1 and W_att2) used in the implementation. We agree this can be misleading. In the revised draft, we will update Eq. (13) to match Algorithm 2 and include both projections explicitly, ensuring full consistency between the mathematical description and the pseudocode. We will also give a detailed explanation here.
>
> Each layer contains three learned matrices:
> W_h: produces a hidden vector (“V-like” projection)
> W_1: produces the first attention-related tensor (“K-like”)
> W_2: produces the second attention-related tensor (“Q-like”)
>
> The input to each layer is the concatenation of: (state, previous layer’s activation at the same time, top-down activation from the next layer, and the action). This gives an input of dimension state_dim + action_dim + hidden_dim + hidden_dim.
>
> W_h projects this input into a tensor of shape hidden_dim. W_1 and W_2 each project this input into a tensor shaped as attention_dim × hidden_dim. These two attention tensors are multiplied together to form an attention matrix of size hidden_dim × hidden_dim.
>
> This attention matrix is then multiplied by the hidden vector from W_h, producing a new vector of size hidden_dim.
>
> Finally, we apply our RMS goodness function to this vector, which collapses it into a single scalar value estimate.
>
> Why this differs from Artificial Dopamine (AD): In AD, the dimensionality of the output of one of the attention projections must match the number of actions. This forces the attention output to match the action-space size and therefore limits how expressive the intermediate computations can be. In our method, the RMS goodness function allows the intermediate vector to have any number of hidden units. This removes the architectural constraint in AD and enables richer nonlinear transformations inside each cell.
> We thank the reviewer again for the helpful feedback. The revised draft now includes the requested clarifications and additional experiments.

---

### Official Review · Reviewer_KK5z · 2025-10-31

**Soundness:** 3
**Presentation:** 3
**Contribution:** 3
**Rating:** 6
**Confidence:** 3

**Summary:**

This paper introduce an ARQ method which does not use the backpropagation update. Experiments on MinAtar benchmarks show ARQ has higher reward than backpropation methods.

**Strengths:**

1. the idea is novel as it bridges FF and TD learning
2. the result looks promising. Experiments on MinAtar benchmarks show ARQ has higher reward than backpropation methods.

**Weaknesses:**

1. The author should carefully review the manuscript for consistency and completeness. For example: Line 156: The term “q-function” should be standardized to “Q-function” for consistency throughout the paper. Line 342: The placeholder “Figure ??” needs to be replaced with the correct figure number.

2. Based on Table 2, while ARQ demonstrates strong performance compared to local RL methods and even surpasses backprop-based algorithms in many discrete tasks, its results on continuous control tasks (DeepMind Control Suite) remain slightly below state-of-the-art actor–critic methods

**Questions:**

1. The authors note that output-layer conditioning is computationally efficient for discrete tasks with low-dimensional action spaces. Could you provide quantitative or qualitative evidence on how ARQ compares in terms of computational efficiency to previous local RL methods (e.g., AD) and backprop-based baselines? For example, does ARQ reduce memory usage, training time, or inference cost?

2. Is it feasible to integrate ARQ into an actor–critic architecture? If so, what are the main challenges or design constraints that would need to be addressed?

3. Could you clarify the conceptual difference between decentralized Q-learning (as in ARQ or multi-agent setups) and distributional Q-learning approaches like D4PG? While distributional Q models the return distribution, does it also imply decentralization, or are these fundamentally distinct paradigms?

4. What are the shades in Figure 5 represents for? is it std or confidence interval? Please be specified in the caption

---

> ### Author Response · Authors · 2025-12-04
>
> We thank the reviewer for the thoughtful feedback and constructive suggestions.
>
> Regarding manuscript consistency and placeholders:
> We appreciate the reviewer pointing out the inconsistencies (e.g., “q-function” vs. “Q-function,” Figure ?? placeholder). We have carefully revised the manuscript to ensure terminology consistency and have corrected all remaining placeholders and formatting issues.
>
> Regarding comparatively weaker DeepMind Control Suite results:
> Historically, backprop-free/local RL methods have struggled to match backprop-based actor–critic algorithms on continuous control tasks. Our goal is therefore not to claim superiority over state-of-the-art backpropagation methods, but to demonstrate that ARQ enables a promising backprop-free reinforcement learning regime. The MinAtar results show that ARQ can outperform both prior local RL methods and even backprop-based baselines in several discrete tasks. While ARQ does not surpass top actor–critic methods on DeepMind Control Suite, we believe these results still indicate meaningful potential for scalable, local Q-learning without gradient backpropagation.
>
> Q1 — Computational efficiency:
> Thank you for the question. In controlled comparisons using the same backbone architecture, ARQ actually trains approximately 6–7× slower than AD (28.94 ms/step vs. 4.51 ms/step). This slowdown is expected and arises from two factors:
>  (1) ARQ has a larger parameter budget due to the ability to use arbitrary dimension in the hidden layer (13.6M vs. 7.27M in our setup) and
>  (2) action conditioning increases the number of samples processed per batch.
> Despite the higher training cost, inference latency remains effectively unchanged—both AD and ARQ exhibit ~1 ms forward-pass latency with similar distributional statistics.
>
> Q2 — Actor–critic integration:
> Integrating ARQ into an actor–critic framework is a compelling future direction. The main challenge is conceptual and architectural: ARQ provides a fully local critic, but it is unclear how to construct an actor that learns without backpropagation while remaining theoretically coherent. Hybrid designs—local critic + backprop-based actor—are feasible, but less aligned with the motivation of purely backprop-free learning. We discuss this limitation and potential research directions in the revised Conclusion section.
>
> Q3 — Decentralized vs. distributional Q-learning:
> These represent orthogonal design dimensions.
>
> Decentralized Q-learning (e.g., ARQ, Seyde et al.) uses multiple independent value-prediction heads—traditionally at the output layer, and in ARQ’s case at every layer—allowing each head to produce its own Q-estimate. This architectural decentralization resembles neuron-level hedonistic learning (Klopf, 1982). Decentralized learning does not aim to capture the spread of the value distribution.
>
> Distributional Q-learning (e.g., D4PG) models the distribution of returns instead of a scalar expectation.
> Distributional learning does not imply decentralization, nor vice-versa. In principle, one could combine both—e.g., decentralized prediction heads that each output a value distribution.
>
> Q4 — Shaded regions in Figure 5
> The shaded regions represent standard deviation across random seeds. We apologize for the ambiguity—this has now been explicitly stated in the caption.

---

### Meta-Review · Area_Chair_Bap3 · 2026-01-07

**Summary:**

The paper proposes ARQ, a method inspired by Forward-Forward (FF) method that avoids backward propagation in the design of Q-network. It is also related to Artificial Dopamine (AD) with similar motivation, but instead of estimate the value of all actions simultaneously, the structure of ARQ allows the value to be conditioned on an input action.

The paper provides a new structure and training method of Q-networks, built on prior work of AD but with meaningful modification that condition every layer with action input.  While the experiments are on simple tasks, they already demonstrate the potential of this line of methods (i.e., local RL without back propagation).  Given the exploratory nature of this work and that all reviewer concerns are well-addressed, I lean on accepting this paper.

**Reviewer Concerns:**

There are several concerns from the reviewers:
- Lack of theoretical justification (xkdR, QJn3) -- addressed:  the authors provide theoretical insight in the rebuttal for the choice of RMS.
- Limited Novelty (QJn3) -- addressed:  The authors clarify that this work does not aim to redesign AD, but to reveal that its learning framework still contains unexplored, high-impact components.
- Limited experiments in terms of number of seeds, scalability, types of ablation studies (xkdR, QJn3) -- addressed:  The authors have added additional seeds in the experiments.  For the other aspects, the authors clarified the scope of the paper and leave large-scale experiments to future work, and also mention that some ablation studies can be complemented by the prior work (AD).

**Reviewer Scores:**

The scores are 6642 in the original review.  The author responses were posted after the openreview incidence and actually at the last minutes of the original discussion phase, so the reviewers won't be able to change their mind even if they fully participate.

---

### Decision · Program_Chairs · 2026-01-26

Accept (Poster)